# Evaluating the Potential Anticancer Properties of *Salvia triloba* in Human-Osteosarcoma U2OS Cell Line and Ovarian Adenocarcinoma SKOV3 Cell Line

Naela Adel Mohammed Saleh [1], Rowan Bahaa El-din Abd El-bary [1], Eric Zadok Mpingirika [1], Hanaa L. Essa [2,3], Mayyada M. H. El-Sayed [2], Mirna Sarkis Sherbetjian [4], Hanin Fadel Elfandi [4], Muhammad Adel Abdel Wahed [4], Rami Arafeh [5] and Asma Amleh [1,4,*]

1 Biotechnology Program, American University in Cairo, New Cairo 11835, Egypt
2 Department of Chemistry, American University in Cairo, New Cairo 11835, Egypt
3 Pesticides Phytotoxicity Department, CAPL, Agriculture Research Centre, Dokki, Giza 12627, Egypt
4 Department of Biology, American University in Cairo, New Cairo 11835, Egypt
5 Palestine-Korea Biotechnology Center, Palestine Polytechnic University, Hebron P.O. Box 198, Palestine
* Correspondence: aamleh@aucegypt.edu

**Abstract:** *Salvia triloba* (*S. triloba*) is an herb inherently linked to traditional medicine systems in the Eastern Mediterranean region. There is minimal experimental evidence however, regarding the anticancer effects of *S. triloba* in both osteosarcoma and ovarian cancer. In this study, we investigated the effects of crude (macerated) *S. triloba* ethanol and acetone leaf extracts on viability, migratory ability, and the expression of genes regulating these activities in U2OS and SKOV3 cells using MTT assay, scratch-wound healing/trans-well migration assay, and RT-qPCR respectively. MTT assay results indicated that the acetone extract significantly reduced both U2OS and SKOV3 cell viability with half-maximal inhibitory concentrations (IC$_{50}$) of 54.51 ± 1.10 μg/mL and 75.96 ± 1.0237 μg/mL respectively; these concentrations further displayed negligible hemolytic activity. The combination of acetone extract (19 μg/mL) and paclitaxel (0.787 μg/mL) displayed synergy and reduced SKOV3 cell viability by over 90%. Additionally, the trans-well migration assay illustrated that the acetone extract (IC$_{50}$) inhibited both U2OS and SKOV3 cell migration by more than 50%. Moreover, *S. triloba* acetone extract significantly downregulated the steady-state mRNA expression of key genes involved in driving select cancer hallmarks. Four fractions were generated from the acetone extract by thin layer chromatography (TLC), and the obtained retention factors (Rf) (ranging from 0.2 to 0.8) suggested a mixture of high and moderately polar compounds whose bioactivities require further investigation. In addition, FTIR measurements of the extract revealed peaks corresponding to OH, aliphatic CH, and ester groups suggesting the presence of phenolic compounds, terpenes, and polysaccharides. Altogether, these results suggest that *S. triloba* possesses potential therapeutic compounds that inhibit cell proliferation and migration, and modulate several genes involved in osteosarcoma and ovarian carcinoma progression.

**Keywords:** *Salvia triloba*; osteosarcoma; ovarian cancer; U2OS; SKOV3; RUNX2; FTIR; TLC-UV fractionation; Chou–Talalay method; drug combination; anticancer

## 1. Introduction

Cancer ranks second among the leading causes of death worldwide and was responsible for approximately 10 million deaths globally in 2020 [1]. It is expected that cancer will become the leading cause of death in this century and that 1 in 5 males and 1 in 6 females will develop cancer during their life time, while 1 in 8 men and 1 in 11 women will die of cancer [2]. Osteosarcoma is the most common type of bone cancer [3], and is prevalent in children and young adults [4,5] where it accounts for approximately 2% of all juvenile

cancers [6]. Furthermore due to its poor prognosis, osteosarcoma is considered the second-greatest contributor to cancer-related mortalities in children [7]. On the other hand, ovarian cancer is the seventh-most-common cancer among females, and one of the primary causes of mortality among gynecologic cancers [8]. Despite the advances made in cancer treatment, approximately 207,000 females die each year due to ovarian cancer [9]. This relatively high mortality rate could be attributed to the fact that ovarian cancer is often diagnosed during advanced stages, since it is usually asymptomatic during early stages [10].

This study utilized both U2OS and SKOV3 cell lines as models for human osteosarcoma and ovarian carcinoma respectively. The U2OS cell line, which was originally retrieved from the tibia of a 15 year old Caucasian female, is moderately differentiated [11], possesses epithelial and adherent morphology, and contains wild-type p53 [12]. In contrast, the SKOV3 cell line was initially isolated from the ovary of a 65-year old Caucasian woman with ovarian adenocarcinoma; the cell line is adherent and possesses epithelial morphology. SKOV3 cells are further reported to be resistant to several chemotherapeutic drugs including cisplatin and doxorubicin [13].

The most commonly employed treatment protocols for osteosarcoma include chemotherapy, radiotherapy, and surgery while ovarian adenocarcinoma is often treated using chemotherapy and surgery [6,14]. However, a major drawback to the utilization of these methods is the development of treatment resistance. It is therefore crucial to continuously develop novel and more effective anticancer therapies. Several plant derived compounds such as taxol, vinca alkaloids (vincristine and vinblastine), and podophyllotoxin analogues were previously approved by the U.S. Food and Drug Administration (FDA) as chemotherapeutic agents; herbal plants thus represent a rich source of various complex compounds such as flavonoids, phenols, alkaloids, lectins, and terpenes that possess potential anticancer effects [15].

Three-lobed sage, *Salvia triloba* L., *syn. S. fruticosa* Mill. (also known as East Mediterranean sage or Greek sage) is a herb native to the East Mediterranean region, ranging from the borders of Southern Italy to Western Syria. *S. triloba* belongs to the largest plant genus, *Salvia*, of the Lamiaceae family, and includes 900 different species distributed all over the world, including Egypt [16,17]. The aerial parts of the plant are commonly boiled and used as a tea infusion to ease stomach ache and other disorders including inflammation and microbial infection [18]. In addition, *S. triloba* was accepted as a natural remedy in both the European and British Pharmacopeia [19,20]. Plants in the genus *Salvia* are reported to contain a variety of potentially therapeutic complex compounds including phenolic acids (such as caffeic acid, rosmarinic acid, and salvianolic acid), and flavonoids (such as luteolin, kaempferol and quercetin). Additionally, the essential oils of this species were found to possess several terpenoids including α and β-thujone, camphor, α-humulene, β-caryophyllene, and 1,8-cineole [21,22]; 1,8-cineole is a major component (>50%) of *S. triloba* essential oils, making *S. triloba* a higher value medicinal plant when compared with other species such as *Salvia officinalis* (*S. officinalis*) [21,22]. These compounds have all been reported to possess a range of bioactivities including antioxidant, anti-inflammatory, anticholinesterase and anti-proliferative effects [21,22].

A limited number of studies have previously examined the anticancer activity of *S. triloba*. *S. triloba* induced both cytotoxicity and apoptosis in prostate cancer (PC-3 and DU-145 cells) [23] and breast cancer (MCF7 and T47D cells) [24,25]. *S. triloba* additionally exhibited potential antiangiogenic activity [26]. The paucity of evidence regarding *S. triloba* anticancer activity warrants further examination of its bioactivity in other cancer types. To the best of our knowledge, *S. triloba* bioactivity in both osteosarcoma and ovarian adenocarcinoma has not been previously tested.

In this study, we examined the potential anticancer effects of *S. triloba* acetone extract on U2OS osteosarcoma cells and SKOV3 ovarian adenocarcinoma cells.

## 2. Materials and Methods

### 2.1. Plant Material Harvest

*S. triloba* leaves were manually harvested from plantations located in Hebron city, Palestine (geographical coordinates: 31° 32′ 4.2216″ N, 35° 6′ 4.302″ E) in spring 2018 and the plant morphological identification was performed and confirmed by Dr. Rami Arafeh through referring to the flora of Syria, Palestine, and Sinai [27].

### 2.2. Crude Extract Preparation

The collected plant material was shade dried at room temperature, then cut into pieces of approximately 1 to 2 cm and ground with an electric grinder into a fine powder. After sieving, 1 g of the powder was mixed with 30.0 mL of either analytical grade ethanol (90%) or total acetone solvent. The mixture was shaken on an orbital shaker (120 rpm) for 24 h at room temperature (25 °C $\pm$ 2), filtered using a glass funnel plugged tightly with cotton, and evaporated in the fume hood (room temperature) until entirely dry. Our choice of extraction solvents was based on the fact that plants in the genus *Salvia* possess a variety of potentially therapeutic phenolic compounds [21] that require the use of moderately polar solvents such as ethanol and acetone to achieve high extraction efficiencies [28]. The obtained yields were 0.0939 g and 0.125 g for acetone and ethanol extracts respectively. The dry extracts were dissolved in dimethyl sulfoxide (DMSO) to make stock solutions of 150 mg/mL. From these stock solutions, a series of working concentrations were prepared by serial dilution using complete cell culture media. To efficiently extract such phenolic compounds, it is recommended to use moderately polar solvents such as ethanol and acetone.

Since the acetone extract showed higher biological activities than the ethanol extract, it was analyzed along with the crude *S. triloba* leaves for their functional groups using Fourier Transform Infrared (FTIR) Spectroscopy. The measurements were conducted on a TGA/FTIR Nicolet 380 spectrometer using 1-mm KBr pellets within the range of 500–4000 cm$^{-1}$.

### 2.3. Cell Culture

The U2OS [29] and SKOV3 [30] cell lines used in this study were respectively provided as gifts from the labaratories of Dr. Andreas Kakarougkas' Lab (Department of Biology, The American University in Cairo, Egypt) and Dr. Anwar Abd Elnaser (Department of Chemistry, The American University in Cairo, Egypt). HepG2 cells were obtained from Nawah Scientific Laboratories in Cairo, Egypt, while HEK-293 [31] cells were obtained as a gift from Dr. Laila Ziko (Department of Biology, The American University in Cairo, Egypt). U2OS and HEK-293 cells were cultured using Dulbecco's Modified Eagle Medium (DMEM) (Invitrogen, USA), while SKOV3 and HepG2 cells were both cultured in Roswell Park Memorial Institute (RPMI) (Invitrogen, USA) basal media. Basal media was supplemented with 10% fetal bovine serum (FBS) (Invitrogen), and 5% Pen-Strep (100 units/mL) (Invitrogen). Cells were incubated in humidified incubators at 37 °C supplied with 5% carbondioxide ($CO_2$).

### 2.4. MTT Viability Assay

The effect of *S. triloba* crude extracts on U2OS, SKOV3, and HepG2 cell viability was evaluated using the 3-(4, 5-dimethylthiazolyl-2)-2, 5-diphenyltetrazolium bromide (MTT) colorimetric assay. Viable cells reduce the yellow MTT reagent (Serva, Germany) reagent to a purple formazan salt, which can be quantified using a spectrophotometer [32]. Cells were seeded at a density of 5000 cells/well in a 96-well plate, and cultured overnight to facilitate attachment. Cell viability was measured after a fixed time point of 48 h. After incubation, the treatment was discarded and replaced by 100 µL of fresh DMEM supplemented with 20% MTT (5 mg/mL) per well and incubated for 4 h. The MTT solution was then discarded and replaced by 100 µL of DMSO (Sigma Aldrich, USA) to dissolve the formazan crystals and the color change quantified using the SPECTROstar-Nano microplate reader (BMG LABTECH), with wavelength parameters set to 570 nm. Cell viability was determined as a percentage of the absorbance of treated cells relative to the absorbance of untreated cells.

HepG2 cells were eliminated from further analysis since treatment with *S. triloba* ethanol extract yielded negligible effect on cell viability. All dose-response curves were generated after transformation and normalization of MTT viability data in GraphPad Prism 6.01 software [33], using Equation (1) (log[inhibitor] vs. normalized response—Variable slope).

$$Y = \frac{100}{\left(1 + 10^{((\text{Log IC50} - X) \times \text{ Hill slope})}\right)} \tag{1}$$

where **X** = log of the concentration of the treatment, **Y** = % viability.

### 2.5. Selectivity Index

The selectivity index (SI) of a cancer treatment is a measure of its cytotoxic selectivity for cancer cells over noncancerous ones, and is obtained by calculating the ratio of the toxic dose of a treatment to its therapeutic dose [34]. In order to calculate the SI of *S. triloba*, we compared the $IC_{50}$ values of both U2OS and SKOV3 cells to that of HEK-293 cells. HEK-293 cells are reported to be derived from human embryonic kidney cells [31], and are frequently used as a normal human cell standard [35–37]. The $IC_{50}$ of *S. triloba* against HEK-293 cells was determined using the MTT assay as previously described, and both cisplatin and paclitaxel were used as positive controls. SI of U2OS and SKOV3 cells were calculated using Equations (2) and (3) respectively.

$$\text{SI (U2OS)} = \frac{\text{IC50 (HEK} - 293)}{\text{IC50 (U2OS)}} \tag{2}$$

$$\text{SI (SKOV3)} = \frac{\text{IC50 (HEK} - 293)}{\text{IC50 (SKOV3)}} \tag{3}$$

where
*SI (U2OS)* is the selectivity index of U2OS cells.
*SI (SKOV3)* is the selectivity index of SKOV3 cells.
*IC50 (HEK-293)* is the $IC_{50}$ of the drug against HEK-293 cells.
*IC50 (U2OS)* is the $IC_{50}$ of the drug against U2OS cells.
*IC50 (SKOV3)* is the $IC_{50}$ of the drug against U2OS cells.

### 2.6. Combination of S. triloba Acetone Extract and Paclitaxel in SKOV3 Cells

We employed the method of Chou and Talalay to explore the cytotoxic effect of *S. triloba* and paclitaxel in combination, and utilized the COMPUSYN software (version 1.0) for results analysis [38–40]. Serial dilutions for each drug, along with the combination were prepared according to the scheme in Table 1. The effect of these concentrations on SKOV3 cells was evaluated using MTT assay. The assay was performed with a seeding density of 5000 cells/well and results taken 48 h post treatment. The COMPUSYN software was utilized to generate combination index (CI) and dose reduction index (DRI) values of the combined treatment according to the median effect principle of Chou and Talalay [38–40].

**Table 1.** Serial dilution scheme for the combination treatment.

| | | \multicolumn{5}{c}{*S. triloba* Acetone Extract} | | | | |
| --- | --- | --- | --- | --- | --- | --- |
| | | $IC_{50} \times 4$ | $IC_{50} \times 2$ | $IC_{50}$ | $IC_{50} \times 0.5$ | $IC_{50} \times 0.25$ |
| **Paclitaxel** | $IC_{50} \times 4$ | Combination 1 | | | | |
| | $IC_{50} \times 2$ | | Combination 2 | | | |
| | $IC_{50}$ | | | Combination 3 | | |
| | $IC_{50} \times 0.5$ | | | | Combination 4 | |
| | $IC_{50} \times 0.25$ | | | | | Combination 5 |

### 2.7. Trypan Blue Exclusion Assay

The trypan blue exclusion assay was employed to provide insight regarding the percentage of non-viable cells after treatment with *S. triloba* extracts and the combination treatment. Cells were mixed with 0.4% *w/v* trypan blue dye (Serva, Heidelberg, Germany) in a ratio of 1:1 and counts for living (unstained) and dead (stained) cells were made using a hemocytometer (Hauser Scientific, USA) and the number of non-viable cells per mL determined using Equation (4) [41]:

$$Non\ viable\ cells\ per\ mL = \frac{number\ of\ nonviable\ cells}{number\ of\ squares\ counted} \times dilution\ factor \times 10,000 \quad (4)$$

### 2.8. Scratch-Wound Healing Assay

The scratch-wound healing assay was utilized to assess the effect of *S. triloba* on cell migration. Cells were seeded (200,000 cells/well) in a 6-well plate and cultured until they reached approximately 70% to 85% confluency. Two perpendicular scratches were made on the cell monolayer using a sterile 200 μL pipette tip and washed twice using $1\times$ PBS. Cells were incubated with respective treatments adjusted to the $IC_{50}$. Pictures of fixed points (22 h for U2OS cells and 20 h for SKOV3 cells) along the scratch were taken using the Olympus IX70 inverted microscope at a series of time points. Image J software (version 1.51j8) was used to measure the wound area, and percentage wound closure was calculated using Equation (5) [42]:

$$WC\ \% = \frac{WC\ 0\ h - WC\ X\ h}{WC\ 0\ h} \times 100 \quad (5)$$

where
    **WC %** is the percentage wound closure.
    **WC 0 h** is the wound area at zero hours.
    **WC X h** is the wound area at a specified time point.

### 2.9. Transwell Migration Assay

The transwell migration assay was further used to examine the effect of *S. triloba* on both U2OS and SKOV3 cell migration. Cells were suspended in 100 μL of culture media supplemented with 1% FBS and applied to the top chamber of a 24-well cell culture insert (8 μm pore size/GBO), at a seeding density of 200,000 cells per well. The lower chamber of the 24-well plate contained 600 μL of culture media supplemented with 10% FBS. U2OS and SKOV3 cells were respectively incubated for 22 h and 10 h with treatments adjusted to $IC_{50}$ values. After incubation, cells in the upper chamber were removed by scraping, while cells that migrated to the lower chamber were fixed using 4% formaldehyde, and stained with 4′,6-diamidino-2-phenylindole (DAPI) (KPL, 71-03-01) (1:1000 in PBS). Fluorescent microscopy was used for the visualization of the stained nuclei at $10\times$ magnification. Pictures of 4 random fields were taken using the Olympus IX70 inverted microscope for all treatments and the number of nuclei per field determined using Image J 1.51j8 software.

### 2.10. Reverse Transcription Quantitative Polymerase Chain Reaction (RT-qPCR)

Reverse transcription quantitative polymerase chain reaction (RT-qPCR) was utilized to analyze differential gene expression between untreated and treated samples. Cells for total RNA extraction were cultured overnight in 6-well plates at a seeding density of 200,000 cells per well. Following overnight culture, the cells were incubated with respective treatments for 48 h. Total RNA was then obtained using Invitrogen™ TRIzol™ Reagent, according to the manufacture's recommendations. After the cells were lysed and homogenized in TRIzol™ Reagent, the homogenate was shaken with chloroform to facilitate phase separation. Total RNA was precipitated from the aqueous layer using isopropanol, and purified by washing in 75% ethanol. Finally the purified RNA resuspended in nuclease free water. Reverse transcription was performed with 0.5 μg of total RNA using RevertAid

First Strand cDNA Synthesis kit (Thermo Scientific, USA) according to the manufacturer's instructions. 1 μL of cDNA (5 ng/μL) was used per reaction. qPCR was performed using PowerUp™ SYBR™ green master mix (ThermoFisher Scientific, Waltham, MA, USA) according to the manufacture's recommendation. GAPDH was used as an endogenous control for all reactions; a list of all genes tested and primers used is summarized in Table S4. The final primer concentration per reaction was 500 nM while the thermo-cycling program used for all genes was 50 °C for 2 min, 95 °C for 2 min, 95 °C for 15 s, and 60 °C for 1 min, for a total of 40 cycles. Annealing temperature for all primers was set to 60 °C. A dissociation step was additionally performed using the following program: 95 °C for 15 s, 60 °C for 1 min, and 95 °C for 15 s. All reactions were performed using the 7500 Real-Time PCR system from Applied Bio-systems, USA, and expression fold changes determined using the $2-\Delta\Delta Ct$ method [43]. Statistical significance was measured using GraphPad Prism 6.01 software's multiple T-tests [33].

### 2.11. Transcription Factor Binding Sites Annotation

Potential RUNX2 transcription factor binding sites (TFBS) were annotated using the Match™ tool within the TRANSFAC® database. The tool searches DNA sequences stored in the library for potential TFBS using positional weight matrices (PWMs) [44]. The parameters used for determining potential transcription factors (TFs) are summarized in Table 2.

**Table 2.** Parameters used for TFBS annotation.

| Parameter | Value |
| --- | --- |
| Matrix library | TRANSFAC MATRIX TABLE, Release 2020.2 |
| Sequence file | MTBP_PM000918140 |
| Profile | vertebrate_non_redundant_minFP.prf |
| Only high-quality matrices | Yes |
| Cut-offs | Minimize false positives |

### 2.12. Hemolysis Assay

The hemolysis assay was utilized to determine *S. triloba*'s hemolytic effect. Institutional Review Board (IRB) approval was granted from the American University in Cairo to collect 2 mL blood samples from three healthy volunteers, after informed consent at the university clinic. Human erythrocytes were exposed to concentrations approximate to either $IC_{50}$ or $IC_{75}$ of *S. triloba* acetone extract. Blood was withdrawn into vials containing Ethylenediaminetetraacetic acid (EDTA) and centrifuged at $1000\times g$ for 5 min, then washed twice using PBS. The Serum was discarded, and 2% erythrocytes were prepared using PBS. 50 μL of the 2% erythrocyte solution was added per well in a round bottom 96-well plate, in addition to 50 μL of each treatment; this resulted into a final erythrocyte concentration of 1% per well. As a negative control (0% hemolysis), 50 μL of PBS was used in place of extract treatment. Deionized water was, on the other hand, used as a positive control (100% hemolysis). Plates were incubated at 37 °C for 1 h, and then centrifuged for 10 min at $3000\times g$ in a plate centrifuge. The resulting supernatant (80 μL) was transferred to a new 96-well plate and absorbance readings were taken at 570 nm. Percentage hemolysis was calculated using Equation (6) [45]:

$$Hemolysis\ (\%) = \frac{(Amax - At)}{(Amax - Amin)} \times 100 \tag{6}$$

where *Amax*, *Amin*, and *At* represent the absorbance values for erythrocytes incubated with deionized water, PBS, and *S. triloba* acetone extract respectively.

### 2.13. TLC-UV Fractionation of Acetone Crude Extract

*S. triloba* acetone crude extract was fractionated using TLC and the separated compounds visualized by ultraviolet light [46]. For fractionation, ready-made TLC plates with silica gel (DC-Fertigfolien Alugram SIL G/UV 254, MACHEREY-NAGEL) were utilized, and the mobile phase was composed of: Ethyl acetate, formic acid, glacial acetic acid, and water in the ratio, 10:1.1:1.1:2.6, respectively. A short wavelength UV lamp (254 nm) was employed to visualize the separated bands/fractions. The RF value of each fraction was calculated by dividing the distance traveled by the fraction over that traveled by the solvent front.

### 2.14. Statistical Analysis

Generated data are presented as mean ± standard deviation of three independent experiments unless otherwise specified. Comparisons between means were performed using either the Student's *t*-test, one-way analysis of variance (ANOVA) or two-way ANOVA and the Dunnett's test was utilized for post hoc analysis. All statistical analysis was performed using GraphPad Prism 6.01 software [33].

## 3. Results

### 3.1. S. triloba Acetone Extract Significantly Reduces U2OS and SKOV3 Cell Viability

The viability of cells exposed to varying doses of *S. triloba* was determined based on absorbance readings obtained from the MTT assay. Readings obtained after 48 h of incubation were normalized to respective negative controls (untreated cells) and expressed as percentage viabilities. *S. triloba* acetone extract significantly reduced U2OS cell viability at all concentrations tested. The highest concentration (250 μg/mL) induced a reduction in viability of more than 50% (Figure 1A). Contrarily, the ethanol extract did not affect U2OS cell viability at all concentrations tested (Figure S1). Cisplatin, which was utilized as the positive control for U2OS cells, significantly inhibited cell viability by more than 50% at all concentrations tested (Figure 1B). SKOV3 cells on the other hand exhibited a reduction in cell viability at only the two highest concentrations tested (75 and 150 μg/mL) when treated with *S. triloba* acetone extract. A drastic decline in SKOV3 cell viability (approximately 82%) was observed when SKOV3 cells were exposed to 150 μg/mL of the acetone extract (Figure 1C). Paclitaxel (SKOV3 positive control) significantly inhibited cell viability at all concentrations tested; reductions in SKOV3 cell viability ranged from approximately 57% to 21% (Figure 1D). Additionally, *S. triloba* ethanol extract showed no effect on HepG2 cell viability when tested with concentrations ranging from 9.4 to 600 μg/mL for both MTT and trypan blue assays. Accordingly, the ethanol extract demonstrated no effect on HepG2 cell morphology at this concentration range (Figure S2). Since HepG2 cell viability displayed no reduction after *S. triloba* treatment, we excluded further testing of HepG2 cells in downstream assays. DMSO, which was the solvent used to reconstitute *S. triloba* extracts exhibted no effect on viability when tested on U2OS or SKOV3 cells at the highest working concentration (0.2% DMSO) (Figure S3).

MTT data were used to determine $IC_{50}$ values by generating dose–response curves for respective treatments. All dose–response curves possessed a negative slope, indicating a reduction in cell viability as the extract concentration increased. $IC_{50}$ values obtained for U2OS were $30.210 \pm 1.180$ μg/mL ($R^2 = 0.797$) and $3.033 \pm 1.189$ μg/mL ($R^2 = 0.919$) for *S. triloba* acetone extract and cisplatin treatments respectively (Figure 2A,B). In contrast, $IC_{50}$ values obtained for SKOV3 cells were $75.960 \pm 1.0237$ μg/mL ($R^2 = 0.890$) and $3.148 \pm 1.197$ μg/mL ($R^2 = 0.771$) for *S. triloba* acetone extract and paclitaxel treatments respectively (Figure 2C,D). Additionally, with the exception of U2OS cells treated with the acetone extract, all other experiments possessed Hill Slope magnitudes greater than 1. The greatest Hill Slope magnitude ($6.678 \pm 3.928$) was observed when SKOV3 cells were treated with *S. triloba* acetone extract while the least magnitude ($0.8665 \pm 0.137$) was recorded for U2OS cells treated with *S. triloba* acetone extract (Table S1).

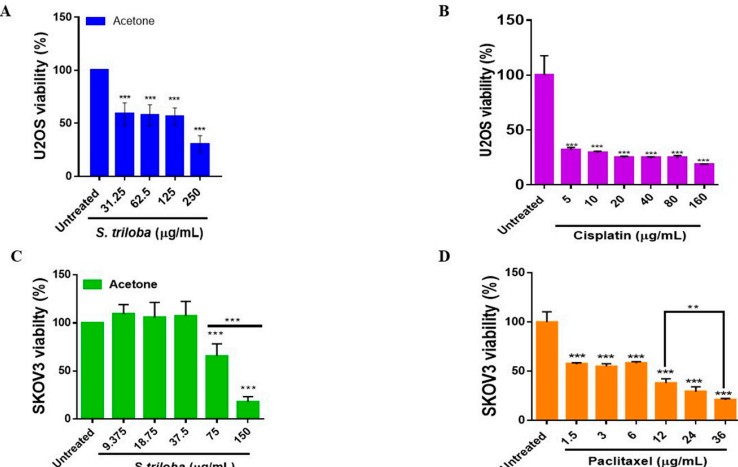

**Figure 1.** *S. triloba* acetone extract significantly reduces U2OS and SKOV3 cell viability. (**A**) U2OS cell viability after 48 h exposure to *S. triloba* acetone. The acetone extract significantly reduced U2OS cell viability at all tested concentrations. (**B**) U2OS cell viability was significantly reduced by more than 50% after incubation with cisplatin (48 h) at all examined concentrations; cisplatin served as a positive control for U2OS cells. (**C**) SKOV3 cell viability was significantly reduced when incubated (48 h) with *S. triloba* acetone extract at both 75 µg/mL and 150 µg/mL concentrations. A sharp decline in SKOV3 cell viability (>50%) was observed at the 150 µg/mL concentration, and this reduction showed statistical significance when compared to the 75 µg/mL concentration. (**D**) Paclitaxel (PTX) served as a positive control for SKOV3 cells; significant reductions in SKOV3 cell viability at all tested concentrations were observed after incubation with PTX (48 h). Comparisons were made between the treated samples and respective untreated controls; results are representative of three independent experiments (*** $p \leq 0.001$, ** $p \leq 0.01$, * $p \leq 0.05$).

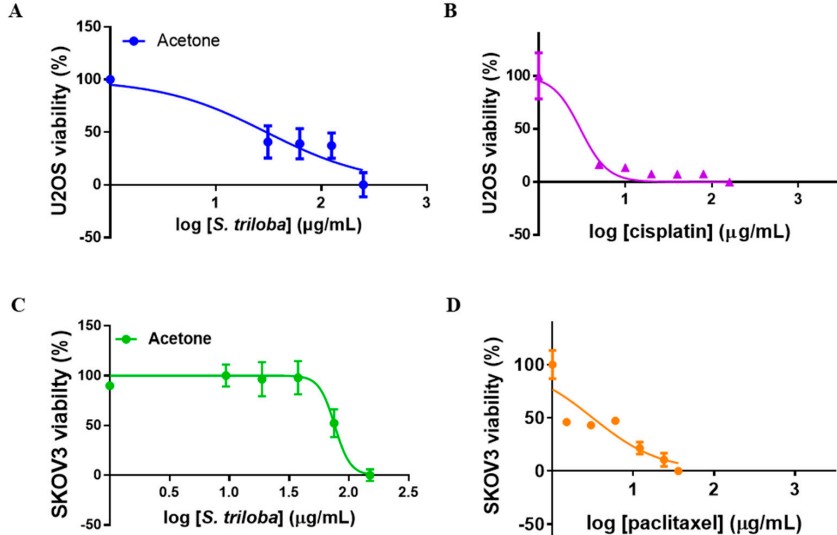

**Figure 2.** U2OS and SKOV3 dose–response curves. All generated dose–response curves had a negative slope, indicating a decrease in cell viability with increasing treatment concentration. The resultant $IC_{50}$ values were (**A**) 30.210 ± 1.180 µg/mL ($R^2$ = 0.797) for U2OS cells treated with *S. triloba* acetone extract, (**B**) 3.033 ± 1.189 µg/mL ($R^2$ = 0.919) for U2OS cells treated with cisplatin, (**C**) 75.960 ± 1.0237 µg/mL ($R^2$ = 0.890) for SKOV3 cells treated with *S. triloba* acetone extract, and (**D**) 3.148 ± 1.197 µg/mL ($R^2$ = 0.771) for SKOV3 cells treated with paclitaxel. Cisplatin and paclitaxel respectively served as positive controls for U2OS and SKOV3 cells; results are representative of three independent experiments.

### 3.2. Selectivity Index of S. triloba Acetone Extract for U2OS and SKOV3 Cells

The selectivity index of *S. triloba* for U2OS and SKOV3 cells was calculated in order to assess the extent to which *S. triloba* selects for cancer cells over noncancerous ones. U2OS cells treated with *S. triloba* displayed the greatest SI (2.51) while SKOV3 cells treated with *S. triloba* had a SI of 1.00. When compared with respective positive controls, the SI (1.167) of U2OS cells treated with cisplatin was about 2-fold lower than the SI of those treated with *S. triloba*; on the other hand, SI (1.316) of cells treated with paclitaxel was slightly higher than the SI of those treated with *S. triloba*. The dose response curves of Hek-293 cells as well as the obtained $IC_{50}$ values are presented in Figure S4 and Table S2 respectively. All calculated SI values are summarized in Table S3.

### 3.3. The Combination of S. triloba Acetone Extract and Paclitaxel Exhibits Strong Synergism in SKOV3 Cells

Drug combination analysis using CompuSyn software facilitated the determination of dose–effect relationships for both *S. triloba* acetone extract and paclitaxel independently, as well as in combination. This analysis aimed to determine whether the combined treatment produces synergistic, additive, or antagonistic effects. Dose–effect data for each treatment independently and in combination indicated a general reduction in SKOV3 cell viability with increasing treatment concentrations (Table 3). A dose-effect plot was generated and used to determine the potency of each treatment after linearization into the median effect plot. The least median effect dose (the dose that results in 50% effect) was observed for paclitaxel (0.221 µg/mL), followed by the combination treatment (13.8357µg/mL), and *S. triloba* (50.928 µg/mL) (Table 4 and Figure 3). By law of mass action, the median effect doses (Dm) were utilized to determine combination indices (CI) for the combined treatment. CI values less than, equal to, and greater than 1 are indicative of synergistic, additive, and antagonistic effects respectively [38,39]. Synergism was observed at only one of the tested combined concentrations (19 µg/mL of *S. triloba* + 0.787 µg/mL of paclitaxel), CI = 0.6415. An over 85% decrease in SKOV3 cell viability was observed for this combination (Table 5 and Figure 4A). The CompuSyn software further simulated dose reduction indices (DRI) at all experimental points tested. DRI values greater than 1 indicate that the dose of a specific drug can be reduced when used in combination. On the other hand, DRIs less than 1 are non-favourable for dose reduction [38,39]. The combination treatment that exhibited synergy had a DRI above 1 for each of the single drug concentrations involved (Table 6 and Figure 4B). The detailed CompySyn report can be accessed in the Supplementary Material (File S1).

**Table 3.** Summary of dose and effect values generated from the combination experiment (SKOV3 cells).

| S. triloba | | Paclitaxel | | Combination ** | |
|---|---|---|---|---|---|
| Dose (µg/mL) | Effect * | Dose (µg/mL) | Effect * | Total Dose (µg/mL) | Effect * |
| 0.25 | 0.01 | 0.01 | 0.01 | 0.2537 | 0.01 |
| 19 | 0.41077 | 0.787 | 0.97536 | 19.787 | 0.88325 |
| 38 | 0.51115 | 1.574 | 0.97752 | 39.574 | 0.85706 |
| 76 | 0.3187 | 3.148 | 0.9 | 79.148 | 0.83451 |
| 152 | 0.65196 | 6.296 | 0.95046 | 158.296 | 0.81048 |
| 304 | 0.9 | 12.592 | 0.97428 | 316.592 | 0.9 |

* Fraction of cells inhibited by treatment; ** *S. triloba* + Paclitaxel.

**Table 4.** Median effect dose for SKOV3 cells.

| Drug | Dm * (µg/mL) | m ** | r *** |
|------|-----------|------|-------|
| *S. triloba* | 50.9282 | 0.85854 | 0.9587 |
| Paclitaxel | 0.22062 | 1.1417 | 0.89187 |
| *S. triloba* + Paclitaxel | 13.8357 | 0.95052 | 0.91799 |

* median effect dose; ** slope of the median-effect (ME) plot; *** linear correlation coefficient of the ME-plot.

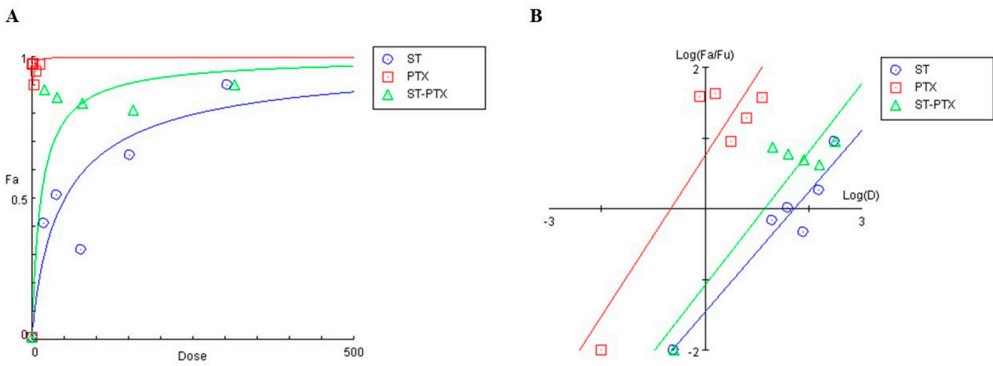

**Figure 3.** Dose–effect curve and median effect plot of the combination experiment in SKOV3 cells. Using dose and effect data obtained from the MTT assay, the CompuSyn software was used to plot the dose–effect curve (**A**); as a prerequisite for determining synergism or antagonism, the dose effect curve is utilized to determine the potency of each drug individually along with the drug combination. The dose–effect plots can be better visualized by linearization in the median effect plot (**B**); the antilog of the x-intercept of the median effect plot gives the median effect dose (Dm) which signifies the potency of each drug (the dose that results in 50% effect). By the law of mass action, the Dm is used to determine the combination index (CI) values. *S. triloba* (ST); paclitaxel (PTX); *S. triloba* + paclitaxel (ST-PTX); fraction of cells inhibited by treatment (Fa); fraction of cells uninhibited by treatment (Fu); dose (D).

**Table 5.** Summary of combination index (CI) values for SKOV3 cells.

| Combination Total Dose (µg/mL) | Effect * | CI Value | CI Description ** |
|-------------------------------|----------|----------|-------------------|
| 0.2537 | 0.01 | 3.5694 | Strong antagonism |
| 19.787 | 0.8833 | 0.6415 | Synergism |
| 39.574 | 0.8571 | 1.5787 | Antagonism |
| 79.148 | 0.8345 | 3.6856 | Strong antagonism |
| 158.296 | 0.8105 | 8.5413 | Strong antagonism |
| 316.592 | 0.9 | 8.7917 | Strong antagonism |

* Fraction of cells inhibited by treatment; ** CI descriptions retrieved from Chou (2006) [39].

**Table 6.** Dose reduction index (DRI) values calculated at experimental points for SKOV3 cells.

| Effect * | DRI *S. triloba* | DRI Paclitaxel |
|----------|------------------|----------------|
| 0.01 | 0.9904 | 0.3907 |
| 0.8833 | 28.3034 | 1.6498 |
| 0.8571 | 10.7945 | 0.6729 |
| 0.8345 | 4.4115 | 0.2891 |
| 0.8105 | 1.8205 | 0.1251 |
| 0.9 | 2.1655 | 0.1201 |

* Fraction of cells inhibited by treatment.

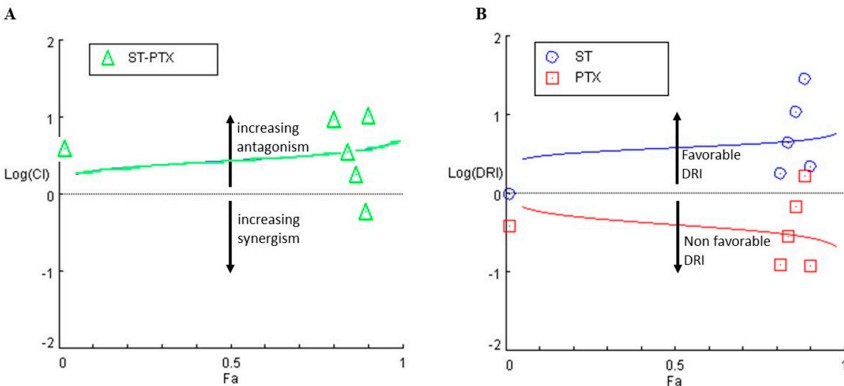

**Figure 4.** Logarithmic combination index plot and Log(DRI) plot for the drug combination experiment in SKOV3 cells. (**A**) A plot of Log(CI) against Fa was used to determine which concentrations of the drug combination exhibited synergy; a CI < 1, CI = 1 and CI > 1 indicates synergism, additive effect and antagonism, respectively. Only one out of six combinations exhibited synergy, all others were antagonistic. (**B**) A plot of Log(DRI) against Fa was used to calculate dose reduction indices (DRI) for the drugs used in the combined treatment. A DRI > 1 indicates that the dose of a given drug is reduced when used in combination than if it were used without combination. DRI values < 1 are not favorable for dose reduction. *S. triloba* (ST); paclitaxel (PTX); *S. triloba* + paclitaxel (ST-PTX); fraction of cells inhibited by treatment (Fa); combination index (CI), dose reduction index (DRI).

### 3.4. S. triloba Acetone Extract Increases the Percentage of Non-Viable Cells in Both U2OS and SKOV3 Cells

Trypan blue exclusion assay was utilized to determine the percentage of non-viable U2OS and SKOV3 cells upon exposure to *S. triloba* acetone extract adjusted to respective $IC_{50}$ values. The percentage of non-viable cells after *S. triloba* treatment increased by (17.758%) and (44.367%) for U2OS and SKOV3 cells respectively; significant increases were, however, noted in SKOV3 cells only when compared with non-treated controls. Furthermore, the combination treatment that exhibited the greatest synergy induced a significantly higher increase (91.967%) in non-viable SKOV3 cells, and this surge was 19.48% greater than that induced by paclitaxel. Both cisplatin ($IC_{50}$) and paclitaxel ($IC_{50}$) served as positive controls for U2OS and SKOV3 cells respectively (Figure 5).

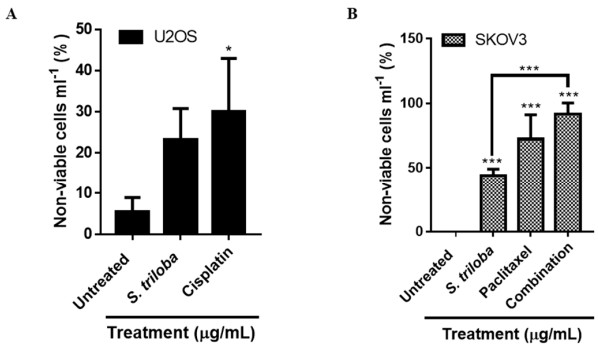

**Figure 5.** *S. triloba* acetone extract increases the percentage of non-viable U2OS and SKOV3 cells. (**A**) When compared with non-treated controls, the number of non-viable U2OS cells increased by 17.8% with no statistical significance. (**B**) Statistically significant results were, however, observed for SKOV3 cells, where the *S. triloba* acetone extract (IC50) along with the combination treatment (*S. triloba* (19 μg/mL) + paclitaxel (0.787 μg/mL)) induced an increase in non-viable cells of about 44% and 92% respectively. Both cisplatin (IC50) and paclitaxel (IC50) served as positive controls for U2OS and SKOV3 cells, respectively, and cells were incubated for 48 h post treatment. Results are a representation of three independent experiments (*** $p \leq 0.001$, ** $p \leq 0.01$, * $p \leq 0.05$).

### 3.5. S. triloba Acetone Extract Decreases Wound Closure in Both U2OS and SKOV3 Cells

The effect of *S. triloba* acetone extract (IC$_{50}$) on U2OS and SKOV3 cell migration was evaluated using the scratch-wound healing assay after incubation for 22 h and 20 h respectively. Wound closure was determined as a percentage of the wound area at either 22 h or 20 h, relative to the wound area at 0 h. Significant decreases in wound closure were noted in both U2OS (31.974%) and SKOV3 (32.881%) cells when compared with respective untreated controls. U2OS cells treated with cisplatin (IC$_{50}$) displayed similar percentage wound closure to cells treated with *S. triloba* acetone extract. On the otherhand, wound closure for SKOV3 cells treated with paclitaxel could not be determined since the treatment caused the cells around the wound boundaries to detach and thus adversely affecting the results (Figure 6).

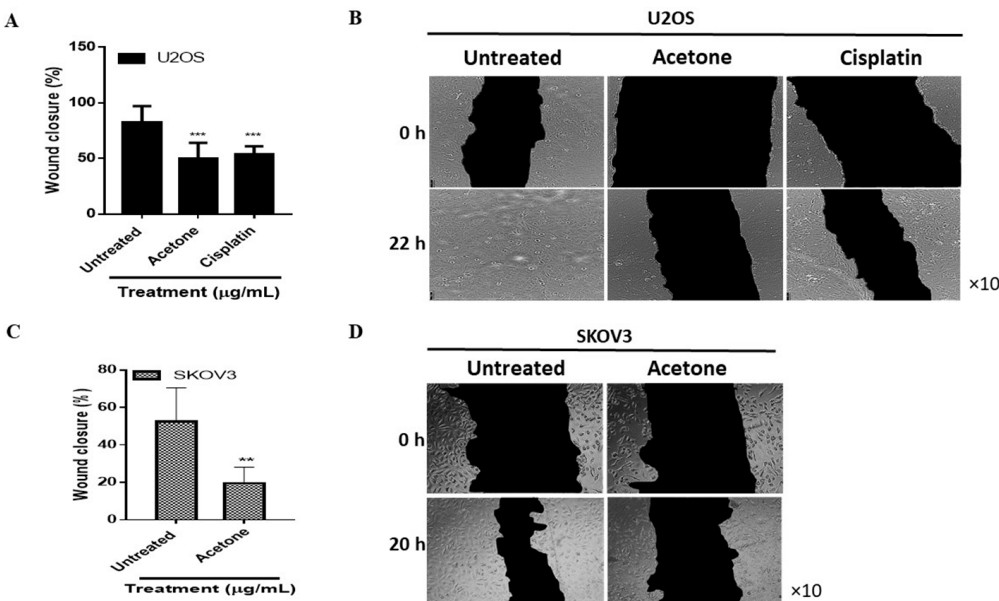

**Figure 6.** *S. triloba* acetone extract decreases wound closure in both U2OS and SKOV3 cells. Percentage wound closure was determined for both U2OS and SKOV3 cells after incubation with *S. triloba* acetone extract (IC$_{50}$) for 22 h and 20 h respectively using the scratch-wound healing assay. Wound closure was determined as a percentage of the wound area at the respective time point relative to the wound area at 0 h. (**A**) *S. triloba* acetone extract significantly decreased U2OS percentage wound closure by about 30% when compared with untreated controls; this decrease was similar to the one observed in the cisplatin control (IC$_{50}$). (**B**) Representative images depicting wound closure progression of U2OS cells. (**C**) Similarly, SKOV3 cells percentage wound closure was significantly decreased by about 32% after *S. triloba* acetone extract treatment. The paclitaxel control (IC$_{50}$) caused the SKOV3 cells around the wound boundaries to detach and percentage wound closure could not be accurately determined for these samples. (**D**) Representative images depicting wound closure progression of SKOV3 cells. Results are a representation of three independent experiments (*** $p \leq 0.001$, ** $p \leq 0.01$, * $p \leq 0.05$).

### 3.6. S. triloba Acetone Extract and the Combination Treatment Respectively Decrease U2OS and SKOV3 Cell Migration

The effect of *S. triloba* on U2OS and SKOV3 cell migration was further assessed using the transwell migration assay. Results from the assay indicated that although *S. triloba* acetone extract significantly decreased U2OS cell migration by 65.6%, SKOV3 cell migrartion was unaffected. Additionally, a 42% decline in SKOV3 cell migration was observed for the combination treatment of *S. triloba* acetone extract (19 µg/mL) and paclitaxel (0.787 µg/mL). The cisplatin (IC$_{50}$) and paclitaxel (IC$_{50}$) positive controls, respectively, displayed negligible effects on U2OS and SKOV3 cell migration (Figure 7).

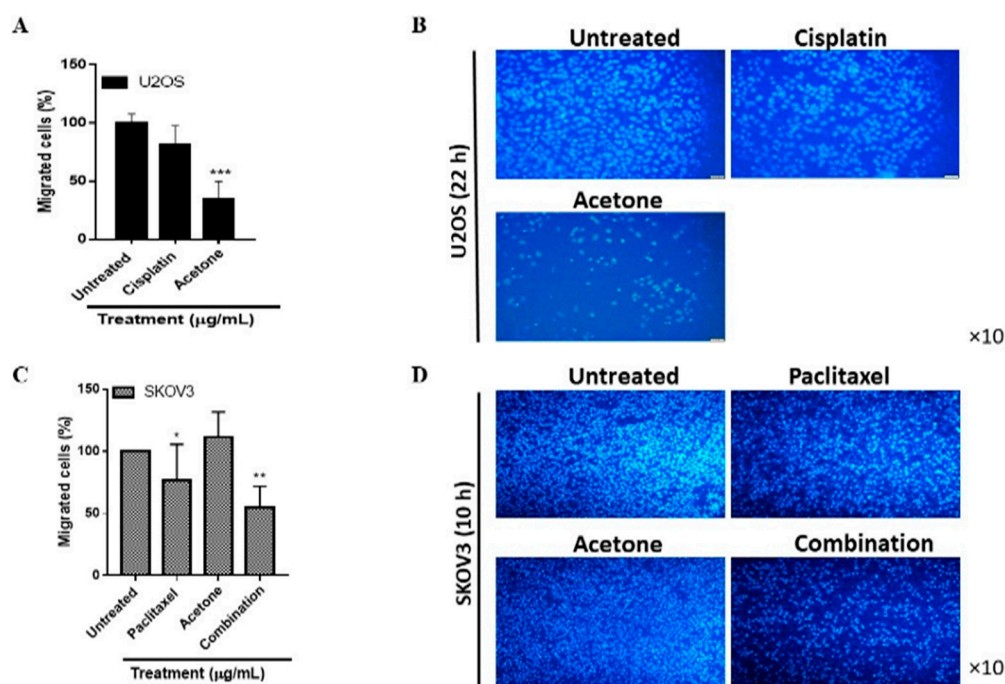

**Figure 7.** *S. triloba* acetone extract decreases U2OS cell migration. The effect of *S. triloba* acetone extract (IC$_{50}$) on U2OS and SKOV3 cell migration was determined using the transwell migration assay and images taken at 22 h and 9 h respectively. (**A**) *S. triloba* acetone extract significantly decreased the percentage of migrated U2OS cells by approximately 70% when compared to untreated controls; nevertheless, the cisplatin (IC$_{50}$) control did not significantly affect U2OS cell migration. (**B**) Representative images displaying DAPI-stained nuclei of U2OS cells that migrated to the lower chamber of the transwell insert. (**C**) On the other hand, SKOV3 cell migration was relatively unaffected after treatment with *S. triloba* acetone extract. However, when the acetone extract was combined with paclitaxel, SKOV3 cell migration was reduced by 42%. (**D**) Representative images displaying DAPI-stained nuclei of SKOV3 cells that migrated to the lower chamber of the transwell insert. Results are a representation of three independent experiments (*** $p \leq 0.001$, ** $p \leq 0.01$, * $p \leq 0.05$).

### 3.7. S. triloba Acetone Extract and Its Combination with Paclitaxel Affect the Steady-State mRNA Expression of Oncogenic Targets in Both U2OS and SKOV3 Cells

The steady-state mRNA expression of several genes involved in cell proliferation, migration, and apoptosis was examined in both U2OS and SKOV3 cells after treatment with either *S. triloba* acetone extract or the combination treatment (*S. triloba* (19 µg/mL) + paclitaxel (0.787 µg/mL)). Comparisons made between treated and untreated U2OS cells revealed significant declines in *RUNX2* ($p \leq 0.001$), *vimentin* ($p \leq 0.001$), *N-cadherin* ($p \leq 0.001$), and *PI3KR1* ($p = 0.013$) expression after *S. triloba* treatment. On the other hand, both *MDM2* ($p \leq 0.001$) and *SETD7* ($p = 0.005$) were significantly upregulated by approximately 1-fold. Expression fold changes in *p53*, *BAX*, and *PTEN* displayed no statistical significance (Figure 8A). Treatment of SKOV3 cells with *S. triloba* acetone extract resulted in a reduced expression of all genes tested; statistically significant reductions were observed for *RUNX2* ($p \leq 0.05$), *BAX* ($p \leq 0.01$), *PTEN* ($p \leq 0.001$), *SETD7* ($p \leq 0.05$), *MDM2* ($p \leq 0.05$), and β-*catenin* ($p \leq 0.01$) (Figure 8B). In contrast, only 3 genes displayed significant changes in expression after testing the combination treatment on SKOV3 cells. Both *RUNX2* ($p \leq 0.05$) and β-*catenin* ($p \leq 0.05$) were significantly downregulated while *p53* ($p \leq 0.050$) displayed an 8.3-fold increase (Figure 8C).

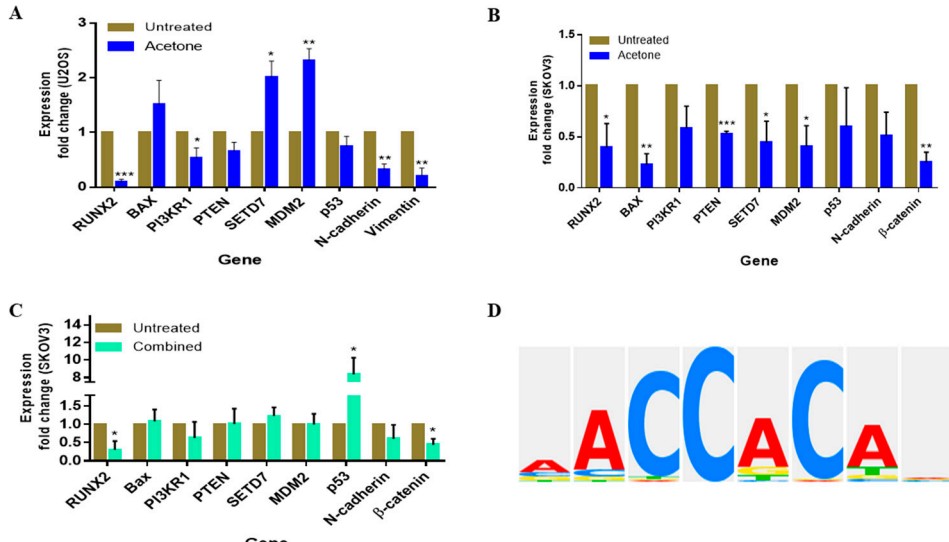

**Figure 8.** *S. triloba* acetone extract and its combination with paclitaxel affect the steady-state mRNA expression of both U2OS and SKOV3 cells. The steady-state mRNA expression of genes involved in cell proliferation and migration was examined using qPCR after cells were treated with *S. triloba* acetone extract ($IC_{50}$) for 48 h. (**A**) Significant decreases in U2OS gene expression were observed for *RUNX2*, *PI3KR1*, *N-cadherin* and *vimentin* genes, while *SETD7* and *MDM2* were significantly upregulated. The greatest increase (approx. 2 fold) in expression was observed for both *SETD7* and *MDM2* genes while *RUNX2* exhibited the highest decline (91%) in expression. (**B**) Statistically significant declines in gene expression were observed for *RUNX2*, *BAX*, *PTEN*, *SETD7*, *MDM2*, and *β-catenin* in SKOV3 cells; none of the tested genes were upregulated after *S. triloba* treatment. (**C**) When SKOV3 cells were treated with the combination treatment (19 μg/mL *S. triloba* + 0.787 μg/mL paclitaxel), *RUNX2* and *β-catenin* expression was significantly downregulated, while *p53* expression was significantly upregulated; *p53* exhibited notable upregulation with an expression fold change of approx. 8. Results are a representation of three independent experiments (*** $p \leq 0.001$, ** $p \leq 0.01$, * $p \leq 0.05$). (**D**) Consensus sequence of RUNX2 transcription factor binding site (TFBS) within the MDM2 promoter region obtained from the TRANSFAC® database.

### 3.8. Identification of RUNX2 Transcription Factor Binding Sites within the MDM2 Gene

In silico examination of MDM2's promoter region predicted a total of six RUNX2 TFBS with core similarity scores ranging from 0.894 to 1 and matrix similarity scores of approximately 0.9. A summary of TFBS start and end positions (relative to transcription start site), along with respective sequences is shown in Table 7. Five out of six of the predicted TFBS sequences possessed the RUNX2 TFBS consensus sequence (AACCACAN) (Figure 8D and Table 7).

**Table 7.** RUNX2 transcription factor binding sites within MDM2 promoter.

| Factor Name | RUNX2 | | | | | |
|---|---|---|---|---|---|---|
| TFBS * | 1 | 2 | 3 | 4 | 5 | 6 |
| Core similarity score | 1.00 | 1.00 | 0.895 | 1.00 | 1.00 | 1.00 |
| Matrix similarity score | 0.998 | 0.918 | 0.906 | 0.894 | 0.893 | 0.913 |
| Start position ** | 68799245 | 68799934 | 68806888 | 68807452 | 68808063 | 68809252 |
| End position ** | 68799254 | 68799943 | 68806897 | 68807461 | 68808072 | 68809261 |
| Sequence | CAACCACAAG | GAACCACTTA | AAGCCACATA | CCACCACGCC | GAGGTGGTGC | TAACCACCTC |
| Reference | ArrayExpress [47] | ArrayExpress [47] | GEO [48] | GEO [49] | ArrayExpress [47] | ArrayExpress [47] |

* Transcription factor binding sites; ** Relative to transcription start site.

### 3.9. S. triloba Acetone Extract Does Not Induce Hemolysis in Human Erythrocytes

Human erythrocytes were subjected to concentrations approximate to either $IC_{50}$ (54.51 µg/mL) or $IC_{75}$ (190 µg/mL) of *S. triloba* acetone extract to examine its hemolytic effect. Deionized water was utilized as a positive control, while untreated erythrocytes were utilized as the negative control. All tested concentrations of the acetone extract induced a negligible hemolytic effect (1.282% and 3.157% respectively for 54.51 µg/mL and 190 µg/mL of the acetone extract) that showed no statistical significance when compared to untreated samples (Figure 9).

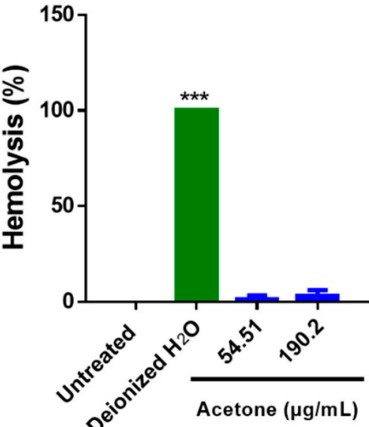

**Figure 9.** *S. triloba* acetone extract does not induce hemolysis in human erythrocytes. Hemolytic activity of *S. triloba* acetone extract at concentrations approximate to either $IC_{50}$ (54.51 µg/mL) or $IC_{75}$ (190 µg/mL) was about 1.282% and 3.157% respectively. When compared to untreated samples, the hemolytic activity of *S. triloba* acetone extract was negligible and the differences were not statistically significant. Deionized water served as a positive control that induced 100% hemolysis. Results are a representation of three independent experiments (*** $p \leq 0.001$, ** $p \leq 0.01$, * $p \leq 0.05$).

### 3.10. TLC-UV Fractionation and FTIR Measurements of Crude S. triloba Acetone Extract

TLC-UV fractionation of the acetone crude extract yielded four fractions (RF1, RF2, RF3, and RF4) with respective retention factors of 0.22, 0.39, 0.57, and 0.81. These varying retention factors indicated that the isolated fractions possessed a range of polarities, with RF1 displaying the least polarity, followed by RF2, RF3, and RF4 (Figure 10 and Table 8).

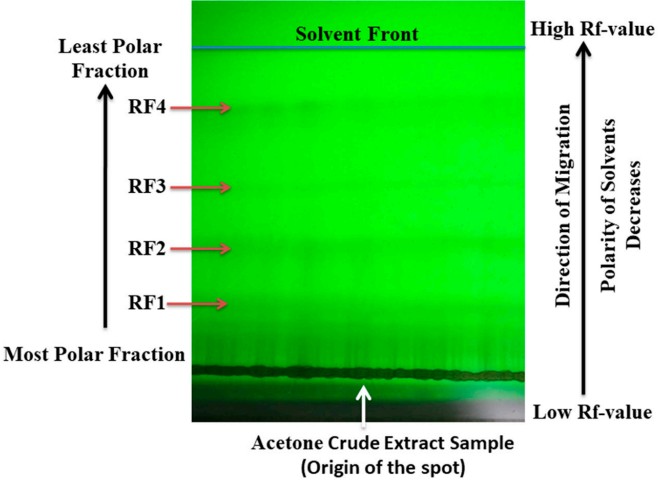

**Figure 10.** TLC-UV fractionation of crude *S. triloba* acetone extract. Four fractions were generated from the acetone extract by thin layer chromatography (TLC); the calculated retention factors (Rf) ranging from 0.2 to 0.8 suggested a mixture of compounds with varying polarities whose bioactivities require further investigation.

**Table 8.** Retention factors of *S. triloba* acetone extract fractionations.

| Distance Traveled by Fraction (cm) | Distance Traveled by Solvent Front (cm) | Retention Factor |
| --- | --- | --- |
| 2.54 | 11.43 | 0.2222 |
| 4.445 | 11.43 | 0.3889 |
| 6.5024 | 11.43 | 0.5689 |
| 9.2202 | 11.43 | 0.8067 |

Furthermore, FTIR measurements were performed on *the S. triliba* leaves and their crude acetone extract, and the obtained spectra are shown in Figure 11. Similar peaks appear in both spectra which indicate that acetone extracted most of the major constituents of the leaves. Peaks at 3420 and 1383 cm$^{-1}$ pertain to the respective stretching and bending of the OH phenolic groups [50,51]. The peaks at 2920 cm$^{-1}$ can be attributed to the stretching vibration of the CH aliphatic groups that probably belong to terpenes [50]. In addition, the peaks corresponding to the carbonyl stretch of the carboxylic acid and ester groups appeared at 1728 and 1262 cm$^{-1}$ [52], while the peaks appearing at 1650 and 1450 cm$^{-1}$ can be ascribed to the vibrations of the uronic acid groups [53,54]. As for the 1109 cm$^{-1}$ peak, it belongs to the acidic polysaccharides [51,55].

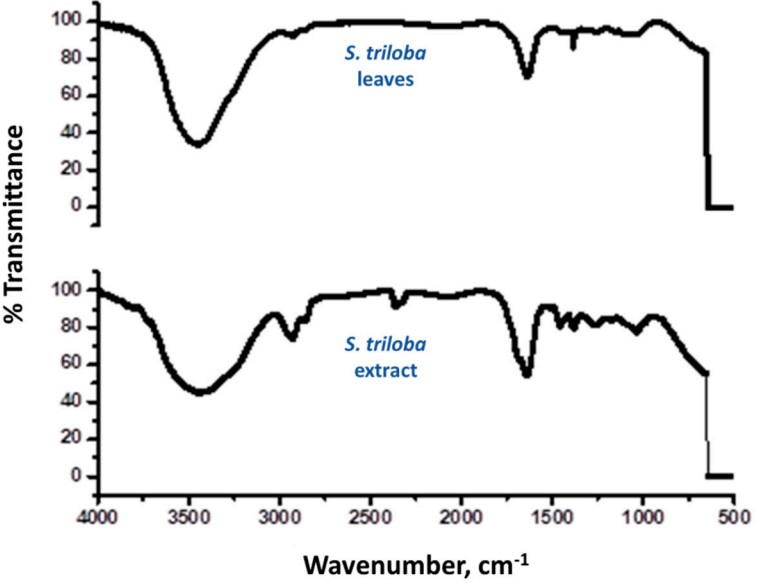

**Figure 11.** FTIR spectra of *S. triloba* leaves (**Top panel**) and its crude acetone extract (**Bottom panel**).

## 4. Discussion

### 4.1. S. triloba Acetone Extract Significantly Reduces U2OS and SKOV3 Cell Viability

In this study, we investigated the effects of crude *S. triloba* ethanol and acetone extracts on viability, migratory ability, and gene expression of various cancer cell lines. The effect of crude *S. triloba* extracts on the viability of U2OS, SKOV3, and HepG2 cells was variable. Although U2OS and SKOV3 cell viability declined significantly after treatment with *S. triloba* acetone extract, HepG2 viability was hardly affected by the *S. triloba* ethanol extract. As a result, HepG2 cells were excluded from further analyses. Although no prior studies have examined the cytotoxicity of *S. triloba* acetone extracts, bioactivities of both ethanolic, and methanolic extracts were previously reported [23,25,56,57]. *S. triloba* ethanolic extracts inhibited the proliferation of MCF7 and TD47 breast cancer cell lines with IC$_{50}$ values of less than 30 μg/mL [57]. When tested on PC-3 and DU-145 prostate cancer cell lines, IC$_{50}$ values of 287 μg/mL, and 456 μg/mL were respectively reported [23]. Our recorded IC$_{50}$ values for *S. triloba* acetone extract against U2OS (30.210 μg/mL) and SKOV3 (75.960 μg/mL)

cells were within the range of previously reported values (17.43 µg/mL to 456 µg/mL) for various cancer cell lines [23,25,56,57]. The observed variability in *S. triloba* IC$_{50}$ can be attributed to differences in the type of cell line used per study. *S. triloba* further exhibited negligible cytotoxicity against the non-tumorigenic mammary cell line, MCF 10A, and such selective cytotoxicity against cancer cell lines supports its potential use in cancer therapy [23]. Treatment of SKOV3 cells with *S. triloba* yielded the greatest Hill Slope (6.678 ± 3.928) while U2OS cells displayed the least Hill Slope (0.8665 ± 0.137). The Hill Slope is equivalent to the Hill coefficient, which gives an idea of binding affinity between target macromolecules in U2OS and SKOV3 cells, and the ligands present in *S. triloba* extract [58]. A Hill coefficient equal to 1 indicates non-cooperative binding while a Hill coefficient either above or below 1, respectively, suggests positive and negative cooperative binding [59]. Our Hill coefficient results thus suggest that SKOV3 cells displayed a greater degree of cooperativity than U2OS cells when treated with *S. triloba*. This distinction could be attributed to differences in target macromolecules between U2OS and SKOV3 cells.

### 4.2. Selectivity Index of S. triloba for U2OS and SKOV3 Cells

Hek-293 cells (normal human embryonic kidney cells) were used to determine the SI of *S. triloba* treatment for both U2OS and SKOV3 cells. Between the two cell lines, *S. triloba* was 2.5 times more selective for U2OS cells than SKOV3 cells. Although *S. triloba* displayed moderate selectivity for U2OS cells, that of SKOV3 was relatively low. This difference in selectivity between U2OS and SKOV3 cell lines could be owed to the fact that the two cell lines are distinct and represent different type of cancer. It is suggested that SI values greater than one are favorable for treatment since they display greater therapeutic effect, with minimal toxicity to non-cancerous cells [60]. The relatively modest SI values observed for both U2OS and SKOV3 cells treated with *S. triloba* could be attributed to the fact that in addition to therapeutic compounds, the crude extract probably contains some toxic compounds, and chemical profiling to identify and remove such compounds is recommended [34]. Moreover, it is argued that selectivity indices can display significant variability due to their dependence on IC$_{50}$ values which could in turn fluctuate based on differences in cell type, cell developmental stages, as well as initial cell densities [61]. Indeed, although Hek-293 cells used for SI calculation in this study are considered a common standard for normal human cells, they are reported to be tumorigenic [31]. Therefore, re-evaluation using a different biosystem(s) it is recommended for herbal drugs with selectivity indices between 1 and 10 [34].

### 4.3. The Combination of S. triloba Acetone Extract and Paclitaxel Exhibits Strong Synergism in SKOV3 Cells

Drug combinations are a foundational aspect of cancer therapy due to their ability to alleviate the toxicity of chemotherapy as well as prevent drug resistance. The use of two or more drugs in combination facilitates the targeting of multiple oncogenic pathways, eventually leading to either synergistic or additive responses. These responses ultimately permit the realization of high therapeutic effect at reduced dose [62–64]. A range of combination studies have examined the synergistic effect of combining plant secondary metabolites with clinical chemotherapeutics [65,66]. When combined with cisplatin, berberine induced both necroptotic and apoptotic cell death in OVCAR3 ovarian cancer cell lines [67]; similar observations were noted when the OV2008 human ovarian cancer cell line and their C13 variant (cisplatin resistant) were treated with a combination of curcumin and either cisplatin or oxaliplatin [68]. Furthermore, the green tea polyphenol derivative, epigallocatechin-3-gallate, re-sensitized 3 ovarian cancer cell lines (SKOV3, CAOV3, and C200 cells) to cisplatin by either 3 or more fold [69]. Since paclitaxel is a plant derived terpenoid [70], we thought it appropriate to have it tested in combination with *S. triloba*. Our expectation was that the combined structural complexity between paclitaxel and *S. triloba* would result in a synergistic effect when tested against SKOV3 cells. In this study, the combination of 19 µg/mL of acetone extract, and 0.787 µg/mL of paclitaxel displayed synergy (CI = 0.6415), and yielded

an inhibition effect of 88.33%. We additionally determined the DRI values for each drug at this inhibition point. In a synergistic combination, the DRI provides a measure of the number of folds by which each drug dose can be reduced at a given effect level. DRI values above 1 indicate that the dose of a given drug can be reduced when used in combination. As such, DRI values less than 1 are unfavorable for dose reduction. Our obtained DRI values thus represented favorable dose reduction folds of approximately 28 and 1.6, for *S. triloba* and paclitaxel respectively [38,39]. These results imply that *S. triloba* contains secondary metabolites that could be utilized for adjuvant therapy [71], when combined with clinical chemotherapeutics.

### 4.4. S. triloba Acetone Extract and Its Combination with Paclitaxel Increase the Percentage of Non-Viable Cells in SKOV3 Cells

We employed the trypan blue assay to determine the percentage of non-viable (dead) U2OS and SKOV3 cells after treatment with either *S. triloba* acetone extract alone, or when in combination with paclitaxel. Although the percentage of non-viable U2OS cells increased by almost 18% after *S. triloba* treatment, the results displayed no statistical significance. Contrarily, the percentage of non-viable SKOV3 cells was significantly greater (44%) after *S. triloba* treatment. The greatest increase in non-viable cells (92%) was, however, observed when *S. triloba* acetone extract and paclitaxel were applied to SKOV3 cells in combination. Since trypan blue dye stains cells with disrupted membranes [41], it is probable that after *S. triloba* treatment, the observed reduction in SKOV3 cell viability in the MTT assay involves the induction of necrotic cell death pathways that induce a rapid loss in membrane integrity [72,73]. The activation of these pathways can be further amplified by the combination treatment. On the other hand, the decreased viability of U2OS cells in the MTT assay probably occur via other cell death pathways that do not lead to rapid losses in cell membrane integrity [72,73]. Besides this, *S. triloba* was reported to induce both apoptotic and necrotic cell death in T47D breast cancer cells through p21 [24].

### 4.5. S. triloba and Its Combination with Paclitaxel Respectively Display Anti-Migratory Properties in U2OS and SKOV3 Cells

Metastasis is the leading cause of cancer-related fatalities, and is a complex process involving cell- migration and invasion [74]. As such, we assessed the anti-migratory potential of *S. triloba* in both U2OS and SKOV3 cells using both the scratch-wound healing assay and the transwell migration assay. The wound healing assay displayed diminished migration of U2OS and SKOV3 cells after *S. triloba* treatment. On the other hand, after treatment with *S. triloba*, only U2OS cells displayed substantial declines in cell migration when the transwell assay was conducted. A significant decrease in SKOV3 cells was only observed for the combination treatment. Several plant derived phenolic compounds were shown to inhibit cancer cell migration by modulating a range of cellular process involved in epithelial-to-mesenchymal transition (EMT), cell invasion, and extravasation [75]. Carnosic acid, an abundant diterpene in *S. triloba* exhibited anti-migratory effects by down-regulating COX-2 in colorectal cancer [76–78]. Carnosic acid further blocked the migration of B16F10 melanoma cells through the inhibition of EMT [77,78]. Our results additionally suggest that the synergistic interaction between *S. triloba* and paclitaxel considerably improves the anti-migratory effect of either treatments. Xiaomeng and colleagues reported a marked increase in SKOV3 cell migration and invasion, along with a down regulation of E-cadherin and MTA3 after combined treatment with β-elemene and paclitaxel [79].

### 4.6. S. triloba and Its Combination with Paclitaxel Modulate the Steady-State mRNA Expression of Genes Involved in Cell Proliferation and Migration

The effect of *S. triloba* acetone extract on the steady-state mRNA expression of several markers implicated in both osteosarcoma and ovarian adenocarcinoma progression was further examined. Since the RUNX2, (PI3K)/AKT, p53, and β-catenin pathways are often abrogated in both osteosarcoma and ovarian adenocarcinoma, we evaluated how *S. triloba* affects the steady-state expression of key genes associated with these pathways.

The RUNX2 transcription factor is a regulator of both osteogenesis and osteogenic differentiation. Normally, RUNX2 acts as a suppressor or promotor of cell proliferation depending on either cell cycle stage or cell type. In osteosarcoma, RUNX2 expression is regularly elevated; moreover, evidence of increased RUNX2 DNA copy number, mRNA, and protein have been reported [80]. RUNX2 is also often upregulated in epithelial ovarian cancer tissue and this overexpression is generally associated with poor prognosis [81]. RUNX2 is further involved in regulating the transcription of multiple luteal related genes in murine ovaries, indicating that it likely plays vital roles in ovarian homeostasis [82]. Overall, RUNX2 is known to modulate diverse pathways such as those involved in cell proliferation, migration, and invasion [80].

p53, another key transcription factor implicated in cancer, serves to halt the proliferation of cells that either possess genetic alterations or are stressed [83]. One pathway by which p53 exerts its anti-proliferation effect is via induction of the intrinsic apoptotic pathway through the pro-apoptotic protein, BAX [84,85]. p53 mediates BAX expression either directly or indirectly through the Bcl-2 family protein Puma [86]. BAX mediates the principal event of the intrinsic apoptotic pathway, which is the release of cytochrome c from the mitochondria into the cytosol, ultimately leading to cell death [87,88]. Additionally, p53 activity is regulated by multiple posttranslational modifications that influence its stability. The principle antagonist of p53 is the E3 ubiquitin ligase, MDM2 that marks p53 for proteasomal degradation. MDM2 thus maintains low p53 levels through an autoregulatory negative feedback loop with p53. Conversely the methyl transferase, SETD7, is reported to regulate various genes including p53. SETD7 promotes p53 stabilization along with inhibiting its nuclear export via p53 methylation [89]. Several studies have suggested that SETD7 methylation is essential for p53 activity, however there is growing evidence that by itself, SETD7 could be insufficient for p53 activation, and that its absence is likely compensated for by other pathways that are yet to be discovered [90].

Similarly, the PI3K/AKT pathway is often over activated in several cancers, including both osteosarcoma and ovarian cancer. The PI3K/AKT pathway is reported to promote cell proliferation, migration, invasion, tumorigenesis, metastasis and chemo-resistance [91,92]. Aberrations in the PI3K/AKT pathway are most likely essential for the development and progression of both osteosarcoma and ovarian cancer, since alterations in the pathway were reported in 100% of advanced-stage osteosarcomas and 70% of ovarian cancer cases [91,92].

Finally, the β-catenin pathway plays a critical role in regulating various cellular processes such as cell fate, polarity, and migration, in addition to neural development, and organogenesis during embryonic development [93]. This pathway is regulated by Wnt glycoproteins which initiate a signaling cascade that eventually leads to phosphorylation of the β-catenin destruction complex and accumulation of unphosphorylated β-catenin. Unphosphorylated β-catenin functions as a coactivator of several downstream target genes when translocated to the nucleus [94,95]. Aberrant Wnt signaling is reported to either initiate or drive the progression of a variety of cancers including colorectal cancer [96], breast cancer [97], hepatocellular carcinoma (HCC) [98], as well as bone [99] and ovarian cancer [100]. β-catenin deregulation is further reported in SKOV3 cells, and mechanisms for deregulation include either mutations in the CTNNB1 gene that encode β-catenin, or mutational inactivation of the β-catenin destruction complex proteins (APC, AXIN1, and AXIN2) [101].

Several plant-based bioactive compounds have been reported to affect gene expression by modulating various interrelated oncogenic transcription factors including RUNX2, p53, and Wnt/B-catenin [102–106]. RUNX2 was implicated in p53 regulation after DNA damage by possibly interacting with HDAC6 to impede p53 mediated pro-apoptotic activity [107,108]. It was further suggested that RUNX2 neutralizes p53 activity through its ability to prevent p53 induced apoptosis [109]. Additionally, stabilization of p53 by inhibiting MDM2 decreased RUNX2 protein expression [110]. RUNX2 activation during osteoblast differentiation was also dependent on p53 inhibition by MDM2 [111]. Both *RUNX2* and *MDM2* were downregulated in SKOV3 cells after *S. triloba* treatment. In

contrast, *MDM2* expression was upregulated in U2OS cells despite *RUNX2* downregulation. Our Transfac database [44] analysis predicted the presence of several RUNX2 transcription factor binding sites within the MDM2 promoter region. The predicted consensus sequence most likely acts as either positive or negative regulatory elements in different cellular contexts. When bound to the consensus AML site (TG[T/C]GGT), both RUNX1 and RUNX2 negatively regulated promoter activity of the human CLC gene [112,113]. However, Makita and colleagues noted that different RUNX2 isoforms can either up- or downregulate their target genes [114] when bound to the consensus RUNX2 site (5′-AACCAC-3′) [115]. Moreover, *S. triloba* did not significantly alter *p53* expression in both SKOV3 and U2OS cells. Abu-Dahab and colleagues reported that *S. triloba* did not affect p53 protein levels in breast cancer cell lines [24]. Hence, it is likely that RUNX2 affects MDM2 activity independent of p53. On the other hand, no RUNX2 transcription factor binding sites were predicted for the SETD7 gene. *SETD7* upregulation after *S. triloba* treatment suggests possible p53 stabilization [90] in U2OS cells. Nevertheless, *SETD7* expression in SKOV3 cells declined after treatment with *S. triloba*. This disparity in *SETD7* expression implies that the regulatory pathways modulating *SETD7* differ from one cellular context to another. Besides, SETD7 was unessential for p53 activation in mouse models [90].

The expression of some genes involved with the RUNX2 and p53 pathways was also assessed after *S. triloba* treatment. *Vimentin* and *N-cadherin* expression was significantly decreased in U2OS cells after treatment with *S. triloba*. Conversely, although *N-cadherin* expression in SKOV3 cells declined, the result was not statistically significant. Vimentin and N-cadherin have been implicated as markers of epithelial–mesenchymal transition (EMT) by modulating cell migration and invasion [48,116]. Generally, N-cadherin promotes both cell migration and invasion in cancer [117], and its expression is correlated with poor survival in ovarian cancer [118]. A few studies however reported that N-cadherin displayed anti-oncogenic properties in both human osteosarcoma tumors and mouse osteosarcoma cell lines where it was downregulated [119]. The downregulation of *N-cadherin* in U2OS cells by *S. triloba* suggests an amelioration of osteosarcoma pathology; nevertheless, the exact role of N-cadherin in osteosarcoma progression remains to be elucidated. Similarly, although the exact role of vimentin in cancer progression and EMT remains unclear, its overexpression is correlated with increased tumor progression, metastasis, invasion and poor prognosis [120]. Finally, the reduced expression of *BAX* in SKOV3 cells infers that the observed decrease in cell viability after *S. triloba* treatment was independent of BAX mediated apoptosis [121,122].

There is growing evidence that both RUNX2 and the PI3K/AKT pathway interact either directly or indirectly to drive tumor progression [123]. On one hand, AKT activity most likely promotes RUNX2 mRNA upregulation, protein stabilization, and transcription activity; while on the other hand, RUNX2 activity stimulates the transcription of *PI3K* and *AKT*, along with components of the mTOR complex [123]. Since *S. triloba* treatment significantly reduced the expression of both *RUNX2* and *PI3K* in U2OS cells, it is possible that *S. triloba* contains bioactive compounds such as phenolic acids, flavonoids, and terpenes [21,22], that either mutually or independently target both RUNX2 and PI3K/AKT signaling. Several isoflavones including genistein, linarin, and hesperetin promoted osteoblast differentiation through RUNX2 pathway activation [124–126], while sulfuretin did so via dual activation of both RUNX2 and AKT signaling [127]. Additionally when tested in osteoperotic rats, the total flavonoids of *Rhizoma drynariae* downregulated RUNX2, but upregulated PI3K, AKT and mTOR protein expression [128]. Intriguingly, *S. triloba* downregulated *PTEN* (inhibitor of PI3K/AKT pathway) expression in SKOV3 cells. As such, the effect of *S. triloba* on the PI3K/AKT pathway most likely varies from one cell line to another. *β-catenin* was also significantly downregulated by *S. triloba*. As observed in *Salvia miltiorrhizae*, *S. triloba* probably increases the ratio of phosphorylated GSK3β while decreasing that of phosphorylated β-catenin [129]. However, making a definitive conclusion necessitates assessing the relative amounts of phosphorylated GSK3β and β-catenin at the protein level.

The observed synergistic effect of the combination treatment could be attributed to its ability to notably upregulate *p53* expression while downregulating *RUNX2* and *β-catenin*. Nonetheless, mutations were reported in the p53 gene for SKOV3 cells, and both p53 transcripts and protein were undetectable [130–132]. Although we identified p53 mRNA in SKOV3 cells, the relatively high cycle threshold values (ranging from 29.791 to 35.983) suggest meager expression [133] in both treated and untreated SKOV3 cells. Additionally, in spite of the fact that paclitaxel induces elevated p53 expression [134], it can trigger p53-independent apoptosis as well [135–137]. Therefore, the combination of *S. triloba* and paclitaxel could have decreased SKOV3 cell viability and migration apart from p53.

*4.7. S. triloba Acetone Extract Does Not Induce Hemolysis in Human Erythrocytes*

Lysis of the erythrocyte membrane is a regular side effect of chemotherapeutic drugs. One of the potential strategies to alleviate chemotherapeutic-induced hemolysis is co-administration of plant derived agents [138]. Several plant based metabolites are potent antioxidants that protect the lipid-rich erythrocyte cell membrane from drug-induced oxidation, and eventual lysis. Phenolics such as caffeic, rosmarinic, and carnosic acids, along with flavonoids such as kaempferol and quercetin possess effective antioxidant potential [138,139]. The abundance of these phytochemicals among *Salvia* species [21] could explain why our extract exhibited negligible hemolytic effect.

*4.8. TLC-UV Fractionation of Crude S. triloba Acetone Extract*

TLC can be used to separate a mixture of compounds based on relative polarity. Compounds that possess lower polarities with respect to the mobile phase will travel much further along the stationary phase than those with higher polarities. Consequently, highly polar compounds display greater Rf values than their moderately polar counterparts [140]. Rf values for the four fractions generated by the *S. triloba* acetone extract ranged from approximately 0.2 to 0.8. This indicates that the extract contained a mixture of both high and moderately polar compounds whose bioactivities and composition require further investigation.

*4.9. FTIR of S. triloba Leaves and Its Crude Acetone Extract*

The similarity in the FTIR spectra of the leaves and its crude acetone extract suggests that they both have the same main functional groups which likely belong to the phenolic compounds, terpenes and polysaccharides. Previous work on *S. triloba* aerial parts collected from El-Sheikh Zoaid in Sinai, Egypt reported the presence of triterpenes, phenolic acids and flavonoids [141]. Several reports on Salvia species such as *S. nemorosa* L., *S. fruticosa*, *S. verticillata*, *S. trichoclada*, *S. suffruticosa*, *S. miltiorrhiza* and *S. officinalis*, and their extracts identified terpenoids, flavonoids and phenolic acids as the major constituents responsible for their potent biological activities [142–147]. Acidic polysaccharides were also obtained from *Salvia officinalis* L. by aqueous extraction followed by ion-exchange chromatography fractionation, and they showed immunomodulatory activity. Prior to fractionation, the polysaccharide extract was confirmed to contain uronic acids along with a mixture of monosaccharides such as arabinose, galactose, and glucose [148].

**5. Conclusions**

As a whole, our results indicate that the crude acetone extract of *S. triloba* posesses anti-proliferative and migratory effects against U2OS and SKOV3 cells, which respectively represent models of osteosarcoma and ovarian carcinoma. These anti-oncogenic effects could be further potentiated by the extract's ability to modulate the steady-state mRNA expession of key targets, such as RUNX2, that promote the progression of both bone and ovarian cancers. Furthermore, co-administering *S. triloba* with paclitaxel may permit appropriate dose reduction of paclitaxel, while potentially alleviating chemotherapy induced hemolysis. The potent biological activity of the extract could be owed to their phenolic, terpenoid, flavonoid, and polysaccharide contents as suggested by the FTIR measurements

along with previous literature reports. To fully elucidate the structure–activity relationships, we recommend further chemical profiling and identification of the bioactive constituents of *S. triloba* extracts, in addition to testing their activity in vivo.

**Supplementary Materials:** The following supporting information can be downloaded at: https://www.mdpi.com/article/10.3390/app122211545/s1, Table S1: Summary of *S. triloba* acetone extract $IC_{50}$ values in U2OS and SKOV3 cells; Table S2: $IC_{50}$ of treatments on Hek-293 cell viability; Table S3: Summary of U2OS and SKOV3 selectivity index values; Table S4: qPCR primer sequences; Figure S1: Effect of *S. triloba* ethanol extract on U2OS cell viability. (A) The *S. triloba* ethanol extract exhibited negligible effect on U2OS cell viability after 48 h exposure. (B) U2OS cell viability was significantly reduced by more than 50% after incubation with cisplatin (48 h) at all examined concentrations; cisplatin served as a positive control for U2OS cells. Figure S2: Effect of *S. triloba* ethanol extract on HepG2 cell viability and morphology; Figure S3: DMSO solvent does not affect U2OS and SKOV3 cell viability; Figure S4: Hek-293 dose–response curves (A) Dose response curve for *S. triloba* acetone extract on Hek293 cells. The percentage viability was normalized to a scale from 0 to 150 and *S. triloba* $IC_{50}$ value was $75.8 \pm 1.060$. (B) Dose–response curve of paclitaxel on Hek-293 cells. The percentage viability was normalized to a scale from 0 to 150 and paclitaxel $IC_{50}$ value was $4.142 \pm 1.132$ μg/mL. (C) Dose–response curve of cisplatin on Hek-293 cells The percentage viability was normalized to a scale from 0 to 150 and cisplatin $IC_{50}$ value was $1.69 \pm 0.868$ μg/mL; File S1: CompySyn report.

**Author Contributions:** Conceptualization, A.A.; formal analysis, N.A.M.S., R.B.E.-d.A.E.-b., E.Z.M., H.L.E., M.M.H.E.-S., M.S.S., H.F.E., M.A.A.W., R.A. and A.A.; funding acquisition, N.A.M.S., R.B.E.-d.A.E.-b., E.Z.M., M.S.S., H.F.E., M.A.A.W. and A.A.; investigation, N.A.M.S., R.B.E.-d.A.E.-b., E.Z.M., H.L.E., M.M.H.E.-S., M.S.S., H.F.E. and M.A.A.W.; methodology, N.A.M.S., R.B.E.-d.A.E.-b., E.Z.M., H.L.E., M.M.H.E.-S., M.S.S., H.F.E., M.A.A.W., R.A. and A.A.; project administration, A.A.; resources, R.A. and A.A.; supervision, A.A.; validation, N.A.M.S., R.B.E.-d.A.E.-b., E.Z.M., H.L.E., M.M.H.E.-S., H.F.E., M.A.A.W., R.A. and A.A.; visualization, N.A.M.S., R.B.E.-d.A.E.-b., E.Z.M., H.L.E., M.M.H.E.-S., M.S.S., H.F.E. and M.A.A.W.; writing—original draft, N.A.M.S., R.B.E.-d.A.E.-b. and E.Z.M.; writing—review and editing, N.A.M.S., R.B.E.-d.A.E.-b., E.Z.M., H.L.E., M.M.H.E.-S., M.S.S., H.F.E., M.A.A.W., R.A. and A.A. All authors have read and agreed to the published version of the manuscript.

**Funding:** This research was funded by Bartlett Foundation Fund for Critical Challenges Grant (grant no. SSE-Bio-A.A-FY19-FY20) for AA and Student Research Grants for N.A.M.S., R.B.E.-d.A.E.-b., M.S.S., H.F.E. and M.A.A.W. from The American University in Cairo.

**Institutional Review Board Statement:** The study was conducted in accordance with the Declaration of Helsinki, and approved by the Institutional Review Board of The American University in Cairo (protocol code 2018-2019-129 and approved on 11 May 2019).

**Informed Consent Statement:** Informed consent was obtained from all subjects involved in the study.

**Data Availability Statement:** The data presented in this study are available both in this article and respective Supplementary Material.

**Acknowledgments:** We thank Ahmed Abdellatif for his critical reading of the manuscript. The graphical abstract was created with BioRender.com.

**Conflicts of Interest:** The authors declare no conflict of interest. The funders had no role in the design of the study; in the collection, analyses, or interpretation of data; in the writing of the manuscript, or in the decision to publish the results.

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
