# Peer review of "Evaluating the Potential Anticancer Properties of Salvia triloba in Human-Osteosarcoma U2OS Cell Line and Ovarian Adenocarcinoma SKOV3 Cell Line"

_applsci, doi:10.3390/app122211545_

Round 1

Reviewer 1 Report

Saleh et al. present interesting results of their study of Salvia triloba extracts in multiple cancer cell line models. These extracts, particularly the acetone, impact cell viability, gene expression, and migratory ability in SKOV3 and U2OS cells. The manuscript is overall well-written and flows logically. However, not all experiments were done in all lines, and it is not clear why this is the case. Further explanations for this and clarification of several other points are needed, and data appears to have been inadvertently duplicated in one table. The paper will be a valuable addition to the literature once these issues are addressed:

Methods:

-Line 160: Equation 1 does not appear to be an equation at all. Please clarify and/or fix any formatting issues.

-In section 2.4, how many cells are seeded for MTT assays? 5,000 cells/well are mentioned in a later section, but not here. Is this still the correct density?

-In section 2.9, please describe how total RNA is obtained for use in qPCR experiments. What kits or reagents were used, and how long after treatment were samples obtained?

-Line 279: Are Amin and Amax reversed? Absorbance would be higher if hemolysis had taken place?

Results:

-Figure 1: was ethanol extract tested on SKOV3? Even if no effect, data for this test would be useful, as it was included for U2OS.

-Similarly, was acetone tested on HepG2? Authors mention no effect of ethanol and that HepG2 was not tested further.

-Table 3 could be placed in supplement, as its key data are already described in the main text.

-Table 4: Paclitaxel and Combination data appear to be duplicated. Please revise with correct data.

-Table 5 requires units.

-Figures 3 and 4 could be combined if the authors so choose.

-How is Table 7 organized? No columns appear in numerical order. Please clarify.

-Figure 5 legend is very nearly what is already stated in main text. Authors may consider revising.

-Figure 5: presumably these data are placed here and not earlier due to the testing of the combination. Why then is there no combination S. triloba and cisplatin treatment for U2OS?

-Authors may consider revising text and Figure 6 legend to emphasize how would healing percentages are calculated (referring to methods).

-In experiments depicted in Figure 6, the authors state they used the IC50 doses of the agents indicated. Why then is cell death only seem with paclitaxel? Images of paclitaxel-treated cells do not appear substantially different from other conditions. Authors should either remove the paclitaxel data in Figures 6C and 6D if cell death really negatively affected results, and comment on lack of toxicity in other conditions.

-Section 3.5: Were transwell experiments done for SKOV3? If so, data should be shown.

-In Figure 8, the authors describe very different gene expression patterns in S. triloba vs. combination treated SKOV3 cells. The doses used are strongly synergistic – are sufficient cells still alive for data collection, or could cell number affect results?

Discussion:

-Line 625: HepG2 again mentioned but what experiments were actually done in this line are unclear (see earlier note).

Reviewer 2 Report

Although the concept of studying the anticancer effects of Salvia triloba are novel and the experiments have been well described and carried out appropriately, however without testing the extracts against normal cell lines, it is unclear whether the effects seen in the sarcoma and ovarian cancer cell lines are cancer specific, or just non-specific toxic effects.  Although reference is made to other studies done with the MCF-10A cell which is a non-tumour forming mammary cell line showing negligible cytotoxicity with S. triloba extracts, studies demonstrating the effects on normal cell lines need to be done concurrently with the studies on the extracts on malignant cell lines.  If there isn’t a “therapeutic window” of effects on the malignant cell lines versus normal cell lines, these extracts are not likely to be of any benefit in the clinic.

Minor comments:

Regarding the statement regarding standard anticancer therapies in sarcoma and ovarian cancer, ineffectiveness of these treatments is really only due to resistance.  The cost of these treatments is not prohibitive, and the side effects are manageable.

Radiotherapy is not routinely used in treatment of ovarian cancer.

Results from studies on the HepG2 hepatocellular cancer cell line are described, however how the studies on this line were performed  are not mentioned in the methods section.

Reviewer 3 Report

After many reflections and doubts, I recommend to reject the submitted manuscript “Evaluating the potential anticancer properties of Salvia triloba in human-osteosarcoma U2OS cell line and ovarian adenocarcinoma SKOV3 cell line” in Applied Sciences. Authors review very interesting topic about the anti-cancer effects of S. triloba in both osteosarcoma and ovarian cancer cell lines, but the article has serious flaws, and some research was not conducted correctly.

Below, I present a list of major and minor concerns:

- Where the cells used in the research (U2OS, SKOV3) came from? From other Lab, ATCC bank or Merck?

- According to the characteristics and handling information, the base medium for both cell lines is McCoy's 5a Medium Modified, not RPMI-1640 or DMEM. Where do such recommendations come from? Such a change of medium may significantly affect the results conducted on the lines, and in addition, the results may be difficult to repeat by other teams that conduct breeding according to the recommendations.

- What concentration of acetone was used, is it total or dilute (how many %)?

- Are the Authors convinced that the acetone concentration used to produce the extract is not toxic to cancer cells? Are you sure it is an effect of S. triloba? I suggest adding an analysis / test (e.g. tumor cell proliferation) with the appropriate concentration of acetone and acetone + S. triloba. If the proliferation decreases, it means the toxic effect of acetone itself (the concentration).

- In graphical abstract it is unreadable why SKOV3 cells were given Paclitaxel and U205 cells were left without stimulation? Is there something missing? Cisplatin? The Fig. 1. Describes the viability of U2OS cells with cisplatin and SKOV3 cells with Paclitaxel.

- Why in Figure 1 there are different doses of S. triloba for U2OS and SKOV3 cells? What was the purpose of this? Why not the same?

- Why - in the Results - Authors describe the effects of S. triloba on other cells - HepG2 cells? (In discussion also). Information that HepG2 cells were excluded by the Authors from further analyses should be included also in Material and Methods. Otherwise the Readers may get lost reading suddenly in the results about another line that the Authors did not mention before. Cells viability, morphology etc. should be tested with concentrations ranging on cells selected for research - U2OS and SKOV3, not HepG2.

- In Hemolysis assay - how many blood samples was used?

- Line 204: Pictures of fixed points along the scratch were taken using the Olympus IX70 inverted microscope at a series of time points – what time points?

- Why in Fig. 6 for both lines the results were shown at different times (20 and 22h)? What lens (x10, x40) was used to take these photos? For the future, please include the scale bars in the pictures.

- In Fig. 4 - what exactly does “*Effect” mean? *Fraction of cells inhibited by treatment – the description of this is illegible…

- Fig.7 and Fig. 9 are too small.

- Self-citation of [44, 45, 47, 48] - I am not convinced that it is properly used at work.

- Inadequate keywords - please consider other, I suggest: salvia triloba, osteosarcoma, ovarian cancer, U-2OS, SK-OV-3, anticancer herbs.

- The template format is shifted. Tables and Figures can be placed centrally, making them easier to read.

- In Authors contribution - please enter initials, not full names and surnames.

The manuscript submitted by the Authors is in line with the subject of the Applied Sciences and could be an attractive article for the Readers, unfortunately it needs a lot of corrections after which Authors may still try to publish their work.

Reviewer 4 Report

The present study reported the effects of S. triloba ethanol and acetone leaf extracts on viability, migration, and the expression of genes in U2OS and SKOV3 cells. While the findings have high merit, some clarifications and revisions are required prior to consideration for acceptance:

1.    Why were ethanolic and acetone extracts used, not aqueous or other solvents? The basis for these two solvents chosen should be clarified.

2.    Plant identification should be done by a botanist.

3.    The cell culture methodology should be elaborated further. For example, the authors tested the extracts on HEPG2 cells, but the methodology did not describe these. The duration in which HEPG2 cells were treated with S. triloba should be detailed.

Furthermore, what is the cell density of U2OS and SKOV3 during seeding for MTT assay?

4.    There are inconsistencies with regard to the experimental design. For example, why the effects of S.triloba and Cisplatin combination was not being explored in any of the analyses? Why the transwell migration assay for SKOV3 was not being included in the study?

5.    Line 508-510 The authors mention "Figure 7 shows the effect of S. triloba acetone extract (IC50) on U2OS cell migration and images were taken at 22 h and 9 h respectively."

Figure 7 (A) and (B) are representatives for which time point? The figures should have shown both 22h and 9h time points respectively. Furthermore, Line 219-220 only mention 22 hours of incubation. The inconsistencies should be clarified.

6.   It is suggested to perform further Pathway analysis based on the available data to demonstrate/visualize the potential molecular process.

7.   Line 844-845 “administering S. triloba along with regular chemotherapy could prevent the occurrence of hemolysis.” The authors did not test S. triloba and drug combination in the hemolysis assay.

8.    Line 880-881 “Furthermore, co-administering S. triloba with cancer drugs, may permit appropriate dose reduction of chemotherapy while alleviating chemotherapy induced hemolysis” The authors only tested the combination of S.triloba with one type of drug (i.e. Paclitaxel). Furthermore, the authors did not test S. triloba and drug combination in the hemolysis assay.

9.    Language and citation should be done meticulously

e.g. Line 881 spelling for chemotherapy

Line 383 Drug combination analysis by CompuSyn software [1] was used…

Line 980-981 JUG-DUJAKOVIĆ, M.; LONÄŒARIĆ, M.; …

Round 2

Reviewer 3 Report

Thank you for revised version of applsci-1857090, but my decision on the "Evaluating the potential anticancer properties of Salvia triloba in human-osteosarcoma U2OS cell line and ovarian adenocarcinoma SKOV3 cell line" was to reject the manuscript , not a reconsider after major revision.
